# Brief Communication: Monitoring snow depth using small, cheap, and easy-to-deploy snow-ground interface temperature sensors

Claire L. Bachand[1,2], Chen Wang[3], Baptiste Dafflon[3], Lauren N. Thomas[1,4], Ian Shirley[3], Sarah Maebius[1,5], Colleen M. Iversen[6], Katrina E. Bennett[1]

[1]Earth and Environmental Sciences, Los Alamos National Laboratory, Los Alamos, NM, USA
[2]University of Alaska Fairbanks, Fairbanks, AK, USA
[3]Earth and Environmental Sciences, Lawrence Berkeley National Laboratory, Berkeley, CA, USA
[4]University of Colorado Boulder, Department of Geography, CO, USA
[5]Princeton University, Department of Civil and Environmental Engineering, Princeton, NJ, USA
[6]Environmental Sciences Division and Climate Change Science Institute, Oak Ridge National Laboratory, Oak Ridge, TN, USA

*Correspondence to*: Claire L. Bachand (clbachand@alaska.edu)

**Abstract.** Temporally continuous snow depth estimates are vital for understanding changing snow patterns and impacts on permafrost in the Arctic. We trained a random forest machine learning model to predict snow depth from variability in snow-ground interface temperature. The model performed well on Alaska's Seward Peninsula where it was trained, and at Arctic evaluation sites (RMSE ≤ 0.15 m). It performed poorly at temperate sites with deeper snowpacks, partially due to training data limitations. Small temperature sensors are cheap and easy to deploy, so this technique enables spatially distributed and temporally continuous snowpack monitoring at high latitudes to an extent previously infeasible.

## 1 Introduction

In the Arctic, snow is an important control on permafrost, as it insulates the ground from cold winter temperatures (Shirley et al., 2022a). Changing snow patterns (Bigalke and Walsh, 2022) and associated ground insulation may accelerate permafrost thaw, leading to the release of large amounts of carbon to the atmosphere (Pedron et al., 2023). Further, changing snow seasonality may alter growing season length and carbon uptake by plants (Shirley et al., 2022b). Snow depth is highly variable at fine spatial scales due to drifting that is affected by topography, vegetation, and wind (Bennett et al., 2022). Drifts form in topographic concavities (e.g., stream beds), while tall shrubs entrap blowing snow. As shrubs expand in the Arctic (Mekonnen et al., 2021), the spatial distribution of snow drifts and subsequent impacts on permafrost may change (Lathrop et al., 2024). Thus, monitoring and modeling fine-scale drifting processes is crucial to understanding permafrost evolution.

These processes are poorly characterized in physics-based models (Crumley et al., 2024), and improvements require robust and fine-scale snow depth validation. However, monitoring the spatio-temporal variability of snow remains a challenge.

Satellite data can be used to estimate snow depth (Besso et al., 2024), but spatial and temporal resolutions are too coarse to
capture the complexity of Arctic snowpacks. End-of-winter snow surveys in remote, high-latitude regions are logistically
difficult but capture the fine-scale spatial distribution of peak snow (Bennett et al., 2022). Machine learning (ML) models can
be used to extrapolate snow survey data, but these estimates still only represent a single point in time (Bennett et al., 2022).
The temporal evolution of snow can be monitored using automated instruments (e.g., snow sonic sensors deployed at Snow
Telemetry (SNOTEL) stations; Fleming et al., 2023), but spatially distributed deployment is time consuming and expensive.
To overcome these challenges, we designed a ML model to extract snow depth from small, inexpensive temperature
sensors located at the snow-ground interface. The model was trained at two small sites on the Seward Peninsula, Alaska, USA,
and evaluated at ten sites distributed across Alaska, Colorado, and New Mexico (USA), Svalbard (Norway), and Siberia
(Russia). Snow dampens temporal variability in snow-ground interface temperature ($T_{SG}$), and thus snow presence/absence
and other snow properties can be identified from $T_{SG}$ data (Lundquist and Lott, 2008; Staub and Delaloye, 2017). Yet, to our
knowledge, this is the first time that a complete time series of snow depth has been extracted from $T_{SG}$ measurements alone.
**2 Methods**
We used data collected at two sites on the Seward Peninsula, Alaska (Fig. A1): 1) A 2.3 km$^2$ gently sloping watershed
located at mile marker 27 along the Nome-Teller Highway near Nome, Alaska (hereafter Teller27) and 2) a 2.5 km$^2$ hillslope
at mile marker 64 of the Nome-Taylor Highway (hereafter Kougarok64). According to end-of-winter snow surveys, the
average peak snow depth from 2017-2019 at Teller27 was 0.96 m, with an average density of 310 kg/m$^3$ (Bennett et al., 2022).
In 2018, snow depth was shallower at Kougarok64 than at Teller27, with an average end-of-winter depth of 0.75 m and density
of 290 kg/m$^3$ (Bennett et al., 2022). Vegetation at Teller27 consisted of mixed sedge-willow-*Dryas* tundra and mixed shrub-
sedge tussock tundra-bog, with some areas of tall willow shrubs (Bennett et al., 2022). Vegetation at Kougarok64 consisted of
tussock-lichen tundra, alder savanna, tall willow shrubs in willow-birch tundra, tall alder shrubs in alder shrublands, and rocky
areas with birch-ericaceous-lichen and sparse *Dryas*-lichen dwarf shrub tundra (Bennett et al., 2022; Breen et al., 2020).
**2.1 Data collection at Teller27 and Kougarok64**
Collocated snow depth and $T_{SG}$ data were obtained at 151 locations across Teller27 and Kougarok64 over the 2021-
2022 snow season via Distributed Temperature Profiling systems (DTPs; locations shown in Fig. A1; conceptual schematic in
Fig. A2) (Dafflon et al., 2022). DTPs were deployed in late September 2021 and snowfall started on October 20, 2021. DTPs
measured temperatures vertically above ground in 5 to 10 cm intervals to a maximum height of 1.67 m. When a sensor is
covered by snow, high-frequency fluctuation of temperature drops dramatically, allowing snow depth to be estimated from
sensor heights. The estimated snow depths have an uncertainty of ± 2.5 cm or ± 5 cm, depending on the sensor spacing. We
estimated $T_{SG}$ from the temperature sensor closest to the snow-ground interface, which ranged from 1 to 5 cm above the ground
surface and thus avoided impacts of soil or moss on the $T_{SG}$ estimate. Additionally, we extracted shallow subsurface
temperature measurements recorded 1 to 5 cm below the ground surface from soil DTPs deployed into the ground (Wang et

al., 2024). 15-minute DTP data were averaged into 4-hour intervals to match the temporal resolution of the miniature temperature sensors described below.

Miniature iButton temperature sensors deployed at the sites (237 total, Fig. A1, A2) recorded $T_{SG}$ from October 1, 2022 to September 18, 2023 in 4-hour intervals. iButtons were placed in vacuum sealed bags and distributed across variable topography and vegetation to capture a broad range of snow conditions. We use the term "tall shrubs" to refer to deciduous shrubs greater than 0.4 m tall with the capacity to reach heights over 2 m (Sulman et al., 2021). Fifty-nine iButtons were placed in tall shrubs (89 outside of tall shrubs) at Teller27, while 41 were placed in tall shrubs (48 outside of tall shrubs) at Kougarok64.

## 2.2 Machine learning model development

Using collocated DTP $T_{SG}$ and snow depth estimates (Sect. 2.1), we developed a random forest ML model to predict snow depth from $T_{SG}$-derived features, which we refer to hereafter as "RF-Seward". We also tested a linear model, a simple neural network, and a Long Short Term Memory (LSTM) model. We chose a random forest as it outperformed or performed similarly to other models. A random forest is simple to design, computationally inexpensive, and easy to interpret. We identified key model features using permutation importance, which reflects how model performance changes when an input feature is randomly shuffled (Breiman, 2001). Larger decreases in performances indicate greater feature importance.

We trained RF-Seward on features derived from the 4-hour DTP $T_{SG}$ data using the hyperparameter values listed in Table B1. For each day, we calculated daily $T_{SG}$ maximum and range. We also considered $T_{SG}$ minimum, mean, and standard deviation, but these features were highly correlated (Pearson's r > 0.9) with other, higher performing, features. To temporally situate RF-Seward (i.e., incorporate information on neighboring snow conditions) and to smooth its predictions, we included daily $T_{SG}$ standard deviations averaged over a 30-day window (length tuned using the validation dataset) prior to, surrounding, and following each day as features in the model. Further, we tested air temperature-derived features, but they did not measurably improve RF-Seward. Ultimately, RF-Seward generated a snow depth prediction for each individual day based on the following $T_{SG}$-derived features (listed in order of permutation feature importance): window-surrounding, window-following, window-prior, daily $T_{SG}$ range, and daily $T_{SG}$ maximum. After finalizing RF-Seward, we retrained the model on all training (96 DTPs) and validation (24 DTPs) data and evaluated its performance on the randomly selected test dataset (31 DTPs). More details on how the training, validation, and test datasets were applied are available in Fig. B1.

Because temperature sensors are often buried under a small layer of soil to protect from direct solar radiation or to monitor soil temperatures (e.g., Lundquist and Lott, 2008), we trained a second ML model, which we refer to as "RF-Below". We used the same hyperparameters and features as RF-Seward, but calculated features from shallow subsurface temperatures measured by 95 soil DTP sensors (76 training and 19 test sensors, locations shown in Fig. A1).

**2.3 Additional model evaluation and application to iButtons**

*Model transferability:* To test model transferability, we trained RF-Seward and RF-Below at Teller27 and tested at Kougarok64, and vice versa. Further, we applied RF-Seward and RF-Below to ten evaluation datasets where $T_{SG}$ and snow depth measurements were collocated (within approximately 5 m of each other). Sites were located in the United States (Alaska, Colorado, and New Mexico), Norway (Svalbard) and Russia (Siberia), with temperature sensors placed at the snow-ground interface or within the top 5 cm of soil (see Table C1). Snow depth was also recorded at the sites (e.g., snow sonic sensors at automated weather stations), and was used to evaluate model performance. End-of-season snowpack bulk densities varied among sites and ranged from 180 kg/m$^3$ (Samoylov Island, Siberia, Russia) to 450 kg/m$^3$ (Senator Beck Basin, CO, USA). Vegetation also varied (Table C3). At Samoylov Island, the temperature sensors were covered by a thick layer of tundra vegetation, while sites in New Mexico, USA vegetation consisted of sparse grasses. Prior to this evaluation, we retrained RF-Seward and RF-Below on all available DTP data (training, validation, and test data).

*Performance in deep snow:* The training data at our study sites was limited to a maximum of 1.77 m due to the length of DTP probes, and thus RF-Seward and RF-Below cannot predict depths greater than 1.77 m. To test if ML could accurately predict deeper snow depths, we trained a third ML model, which we refer to as "RF-Deep". To train this model, we supplemented our original Seward Peninsula training dataset with additional data from two model evaluation sites in Senator Beck Basin, CO, USA with deeper snowpacks (Table C1). The model was applied to one site and trained with data from the other (in addition to the Seward Peninsula DTP data). To mimic the distribution of snow depths at these sites, we ensured that 10 % of the training data consisted of snow depths above 2 m. This reduced the training dataset size compared to other models (Table B2).

*Model application to iButtons:* We applied RF-Seward to iButtons deployed at Teller27 and Kougarok64 (Sect. 2.1) to assess how tall shrubs affect snow depth and $T_{SG}$. We divided the iButtons into two groups: within and outside of tall shrubs. We averaged $T_{SG}$ measurements and snow depth predictions over a period corresponding to peak snow (March 20-April 9). We used the non-parametric Wilcoxon rank-sum test (Wilcoxon, 1945) to identify statistical differences in snow and $T_{SG}$ conditions between shrubs and no-shrubs.

**3 Results and discussion**

RF-Seward performed well on the test dataset ($R^2 = 0.87$; RMSE = 0.15 m; mean bias = 0.03 m; Fig. 1a, g), but underestimated snow depths when trained at Teller27 and tested at Kougarok64 ($R^2 = 0.85$; RMSE = 0.17 m; mean bias = -0.10 m; Fig. 1b) and overestimated when trained at Kougarok64 and tested at Teller27 ($R^2 = 0.72$; RMSE = 0.23 m; mean bias = 0.05 m; Fig 1c). Differing air temperature regimes between Teller27 (warmer) and Kougarok64 (colder) may have contributed to these biases (i.e. same snow depth at the two locations corresponded to different $T_{SG}$). However, all RF-Seward features were derived from $T_{SG}$ variability (not magnitudes), except for $T_{SG}$ maximum. Excluding $T_{SG}$ maximum from the model (not shown) did not eliminate the biases seen in Fig. 1c, d, suggesting that these errors may be tied to factors that affect

T$_{SG}$ ranges (e.g., latent heat processes). RF-Below performed worse than RF-Seward and did not transfer as well between sites
(Fig. 1d – f, h, i), likely due to variability in ground insulation properties (i.e. soil type, vegetation, etc.) which confound the
snow insulation effect. Further, warmer and/or wetter sites (e.g., Teller27) undergo more freezing and thawing than colder
and/or dryer sites (e.g., Kougarok64), producing zero-curtain periods where the key snow depth predictor (temperature
variability) flattens at 0°C as water changes phase (Staub and Delaloye, 2017).

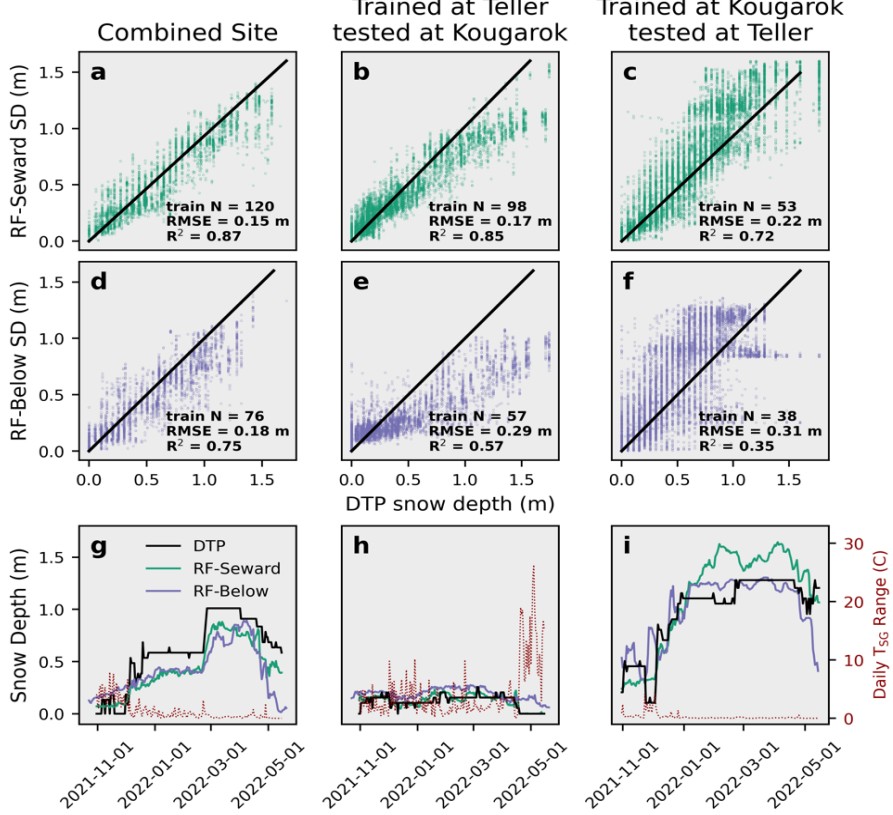

**Figure 1. Performance of RF-Seward a) evaluated using test data, b) when trained at Teller27 and tested at Kougarok64, and c)**
**visa versa. d-f) Same as a-c but for RF-Below. Time series plots of DTP snow depth data vs. ML estimates when g) trained at both**
**sites, h) trained at Teller2, and i) trained at Kougarok64. The dotted red line shows daily T$_{SG}$ range, with narrower temperature**
**ranges occurring under deeper snow cover. "Train N" refers to the number of DTP sensors used to train each model.**
RF-Seward performed well at the two sites where T$_{SG}$ data were available in the Arctic: Bayelva station in Norway
(RMSE = 0.15 m; mean bias = 0.02 m; Fig. 2a) and Imnavait Creek, on Alaska's North Slope (RMSE = 0.08 m; mean bias =
-0.04 m; Fig. 2b), indicating that the model may be transferable to other pan-Arctic locations. Additionally, we tested RF-
Seward and RF-Below at four sites in the Arctic where temperature was recorded below the ground surface. At Samoylov
Island, Russia (Fig. 2e), sensors were placed below an insulating layer of wet tundra vegetation, which caused RF-Seward to

overpredict snow depth (mean bias = 0.40 m). RF-Below decreased overestimations at Samoylov Island (mean bias = 0.14 m) and at other sites in Alaska (Figure 2 c,d,f). RF-Below performed best at sites near Council on the Seward Peninsula, Alaska, USA, likely because vegetation at these sites is most similar to vegetation at the training study sites.

In New Mexico, USA, paired iButtons recorded above and below ground temperature data at two sites (A and B). Predictions from iButtons placed above the ground surface were averaged into a single RF-Seward estimate, while predictions from iButtons placed below the ground surface were averaged into a single RF-Below estimate. At Site B, RF-Seward and RF-Below underpredicted peak snow by about 0.07 m (Fig. 2g). RF-Seward performed better at Site A (observed peak snow = 0.18 m; predicted = 0.16 m), although RF-Below still underpredicted by 0.10 m, possibly because the model expected insulating tundra vegetation. Both models performed worse when applied in the wrong context (i.e. RF-Seward overpredicted peak snow by 0.13 m when applied to below ground data; RF-Below underpredicted peak snow by 0.16 m when applied to above ground data), indicating that excess insulation from a thin layer of soil or vegetation will be confused for snow.

Performance at the New Mexico, USA sites fell within RF-Seward and RF-Below's typical ranges, despite the higher end-of-season bulk density compared to Arctic snow (~ 400 kg/m$^3$ vs. 300 kg/m$^3$). However, zero-curtain periods (ZCPs) caused the model to occasionally overestimate snow depth. For the above ground iButtons, ZCPs were likely caused by water pooling and freezing on top of the iButton's vacuum-sealed bag and by water freezing at the bottom of the snowpack following rain-on-snow (ROS; Staub and Delaloye, 2017). In New Mexico, USA, ROS occurred from January 21 – 25, 2024, leading to an erroneous snow accumulation event in Fig. 2g. ZCPs were more prevalent in the below ground data due to the repetitive freeze-thaw of the soils during snow-free periods of the winter, causing erroneous RF-Below predictions (e.g., early snow accumulation in Fig. 2g.). Our results suggest that RF-Below will perform poorly for warm, ephemeral snowpacks, which are expected to become more common as the climate warms (Wieder et al., 2022). ZCPs completely dampen $T_{SG}$ variability and therefore uncouple $T_{SG}$ from snow depth. Even given training data more representative of ZCPs, snow depth estimates may remain unreliable during these periods. Incorporating features into the model which indicate the presence of ZCPs may reduce these errors. Further, deploying iButtons at the snow-ground interface (rather than below ground) decreases the number of ZCPs in the temperature data.

Below ground temperature data was recorded at Grand Mesa, Colorado, USA (Fig. 2h), while $T_{SG}$ was recorded at two sites in Senator Beck Basin, Colorado, USA (Fig. 2i-j). These sites accumulated more snow (up to 2.85 m) than the sites where RF-Seward was trained (maximum depth = 1.77 m), resulting in underpredictions of deep snow at these sites (Fig. 2h-j). RF-Deep predicted deeper snow depths than RF-Seward, although predictions still leveled off prematurely for some years (e.g., 2008 - 2009 in Fig. 2j). RF-Deep also appeared visually noisy compared to RF-Seward, possibly due to the smaller training dataset (Table B2) and lower quality training data (i.e., temperature and snow depth measurements were not perfectly collocated). RF-Deep's poor performance indicates that at a certain depth, $T_{SG}$ may be dampened to the extent that ML can no longer accurately predict snow depth. Past research has shown that snow depths greater than 0.5 m can completely insulate the ground, although even snowpacks deeper than 4 m are not always fully insulating (Slater et al., 2017; Staub and Delaloye, 2017, their Fig. 5). Because of this, it is likely that deep snow decreases the predictive value of $T_{SG}$ measurements, which will

1  have a minimal effect on understanding soil temperature but could cause major errors when estimating water availability from

2  snow depth predictions.

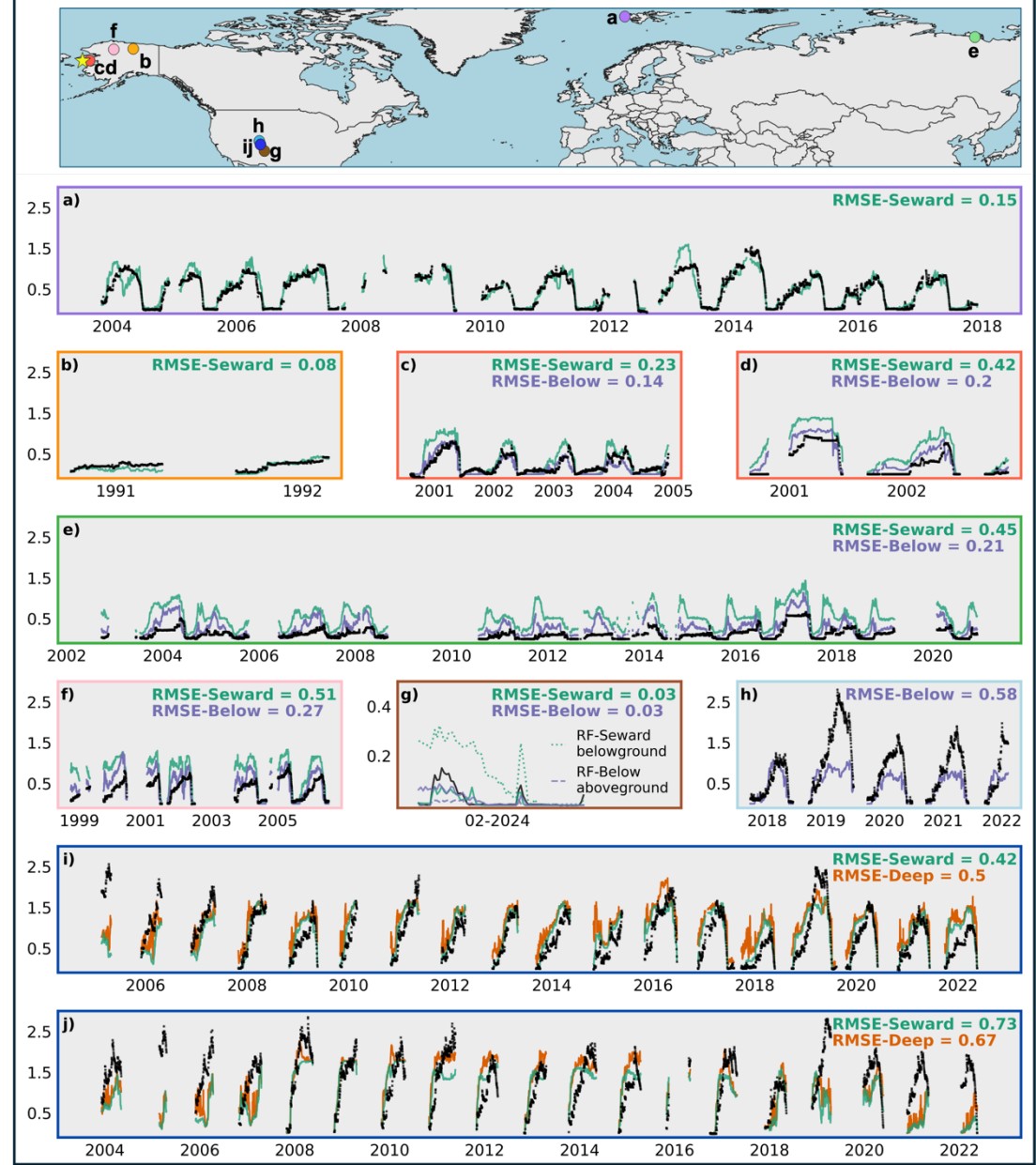

**Figure 2. ML performance at a) Bayelva Station, Svalbard, Norway (Boike et al., 2017, 2018); b) Imnaviat Creek, Alaska, USA (Sturm and Holmgren, 1994; Stuefer et al., 2020); c,d) Council, Alaska, USA (Hinzman et al., 2016); e) Samoylov Island, Siberia, Russia (Boike et al., 2019a,b); f) Ivotuk, Alaska, USA (Hinzman et al., 2016); g) Los Alamos, New Mexico, USA (Thomas et al., 2024); h) Grand Mesa, Colorado, USA (Houser et al., 2022); and i,j) Senator Beck Basin, Colorado, USA (Center for Snow and Avalanche Studies, 2012; Landry et al., 2014). Locations are shown on a map, with the yellow star indicating the Seward Peninsula of Alaska,**

**USA, where RF-Seward was trained. Black lines show measured snow depth at each site. Y-axis and RMSE values indicate snow**
**depth in meters. Note adjusted y-axis for Los Alamos, New Mexico, USA (g). For this site, we also show RF-Seward and RF-Below**
**predictions when RF-Below was applied above ground and RF-Seward was applied below ground (dotted lines).**
*Model application:* Shrubs can entrain blowing snow, resulting in snow drifts (Bennett et al., 2022). Averaged from
March 20th-April 9th 2023, the ML model estimated 0.33 m more snow for iButtons deployed in shrubs than outside of shrubs
($p < 0.001$). This result may be biased low as RF-Seward rarely predicted more than 1.5 m of snow due to training data
limitations. $T_{SG}$ averaged from March 20th to April 9th was 1.65 °C warmer in tall shrubs than outside of tall shrubs ($p < 0.001$).
This provides evidence that increasing arctic shrubification (Mekonnen et al., 2021) may increase snow depths, insulate the
subsurface in winter, and accelerate permafrost thaw as suggested by Sturm et al. (2001). However, topographic and landscape
characteristics can drive the formation of deep snow drifts even without the presence of tall shrubs (Parr et al., 2020). The
iButton with the third highest snow depth prediction, averaged from March 20th to April 9th (1.46 m), was placed in short
grasses adjacent to a stream bed, which likely experienced snow drifting due to topographic concavity (Parr et al., 2020).
Similarly, the iButton with the fourth highest snow depth prediction (1.45 m) was placed near the edge of dense tall shrubs,
where snow may have also accumulated (Currier and Lundquist, 2018).
**4 Conclusions**
We trained a ML model to predict snow depth from variability in snow-ground interface temperature. The model
performed well on the test dataset and at two arctic evaluation sites (RMSE <= 0.15 m). Small temperature sensors are cheap
and easy-to-deploy, so this technique enabled spatially distributed and temporally continuous snowpack monitoring to an
extent previously infeasible. Additional collocated $T_{SG}$ and snow depth observations could be used to retrain the model and
enhance its transferability. While the model generally performed well, rain-on-snow and zero-curtain periods caused the model
to erroneously predict snow accumulation events. Further, the model failed to replicate deep snow (greater than 1.5 m) observed
in Colorado, USA. For optimal performance, the model should be applied to temperatures recorded at the snow-ground
interface. Predictions made using subsurface temperatures were impacted by varying soil types, vegetation properties, and
latent heat processes. Using ML predictions, we found that snow at Teller27 and Kougarok64 was significantly deeper in
patches of tall shrubs than outside of tall shrubs, and that $T_{SG}$ averaged from March 20th to April 9th was on average 1.65 °C
warmer within tall shrubs.
Future research should focus on developing this technique for locations where peak snow depths exceed 1.5 m (e.g.,
Colorado, USA), as these regions are crucial for water security across the world.  While deep snow may completely dampen
$T_{SG}$, it is possible that the ML model will perform better given a larger and more representative training dataset and/or
additional input features. Alternatively, this technique could be combined with other monitoring and/or modeling efforts. For
example, snow depth estimates made early in the snow season (e.g., when snow is shallow) could be used to estimate snow

variability across the landscape and to downscale coarse model or remote sensing snow depth estimates. Further, the application of a ML model tailored towards time series estimates (e.g., a Long Short Term Memory Model; LSTM) could improve predictions. In this study, we only had one year of data, which likely limited the LSTM's performance. With a longer-term dataset, we could provide the LSTM with more training points and a longer look-back window (e.g., an entire snow season), which would likely enhance its performance. Additionally, how snow stratigraphy and density affect model results remains unclear. The sites examined here typically experienced frozen soil prior to snowmelt, and therefore, how unfrozen soils affect ML predictions should also be explored.

*Code/Data Availability:* Snow depth predictions are available on the Environmental System Science Data Infrastructure for a Virtual Ecosystem (ESS-DIVE) data portal (Bachand et al., 2024; https://doi.org/10.15485/2371854). The data package includes a *.csv file of RF-Seward and RF-Below predictions at sites in the United States (Alaska, Colorado, and New Mexico) as well as Norway and Siberia, Russia. The machine learning model is available on Github (https://github.com/cbachand-LANL/iButton-SnowDepth-ML).  The code package includes a *.joblib file of the trained random forest models, which can be downloaded and directly applied to new datasets. Example workflows for cleaning data inputs, training machine learning models, and making predictions are also included in a *ipynb file. iButton temperature measurements at Teller27 and Kougarok64 (Bennett et al., 2024; https://doi.org/10.15485/2319246) and at the Los Alamos, New Mexico, USA study sites (Thomas et al., 2024; https://doi.org/10.15485/2338028) are available on ESS-DIVE, as well as the DTPs temperature and snow depth data used in this study (https://doi.org/10.15485/2475020).

*Author contributions:* CLB wrote the manuscript draft, developed random forest methodology, and performed analysis; CW developed methodology to estimate snow depth using DTPs and curated data; BD, CW and IS led the DTP deployment, data collection and analysis; LNT led iButton data collection campaigns in Los Alamos, NM, USA; SM developed LSTM methodology; CMI acquired funding and is the PI of the NGEE Arctic project; KEB developed the data collection study at Teller27 and Kougarok64, supervised research, developed the research concept, contributed original text, and is the Institutional Lead of the NGEE Arctic project at LANL; all authors reviewed and edited the manuscript.

*Competing interests:* **The authors declare that they have no conflict of interest.**

*Acknowledgements:* We gratefully acknowledge the Mary's Igloo (Qawiaraq in Iñupiaq), Sitnasuak, and Council Native Corporations for guidance and for allowing us to conduct our research on their traditional lands. The authors gratefully acknowledge the contributions of Shannon Dillard, Ryan Crumley, Eve Gasarch, Evan Thaler, Jerome Quintana, Kenneth Waight, Stijn Wielandt, John Lamb, Sylvain Fiolleau, Sebastian Uhlemann and Craig Ulrich for assisting in data collection at the Teller27, Kougarok64, and Los Alamos, New Mexico, USA study sites. We thank Mia Mitchell for her assistance with

Fig. A1. Further, we thank Julia Boike and Mathew Sturm for providing insight and data sources as we developed this machine learning approach.

*Financial Support:* The Next Generation Ecosystem Experiment in the Arctic (NGEE Arctic) project is supported by the Office of Biological and Environmental Research in the U.S. Department of Energy's Office of Science.

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
