# Peer review of "Brief Communication: Monitoring snow depth using small, cheap,"

_EGUsphere, 2024_

## Author Comment (AC1)

**Reviewer 1**

Summary & General Comments:

This brief communication presents a novel approach to derive snow depth from low-cost temperature sensors deployed at the snow-ground interface using a random forest model. This method would hypothetically allow snow depth monitoring at far greater number of sites than currently available, a significant finding well within the remit of The Cryosphere journal. The manuscript is concise and well-written. I recommend this manuscript be published subject to minor revisions, as detailed below.

Thank you for your review of our manuscript. We appreciate your thoughtful feedback.

As an aside, I would be interested to see what you think the impact of snow stratigraphy may be on the depth estimates from your model (particularly when it comes to evaluating snow depth at sites beyond the Arctic, such as the New Mexico site), but I understand that you are unlikely to have this data for comparison.

This is an interesting question which we have not yet explored. I would think that the accuracy of the machine learning algorithm may decrease at locations where the stratigraphy (and more generally, snowpack physical properties) is different from where the model has been trained, but we do not have the data to test this. We have added a mention of snow stratigraphy to the conclusions to address this area of uncertainty:

P10-L[3]: ***"Additionally, how snow stratigraphy and density affect model results remains unclear ."***

Minor/Technical Comments:

This is a broad and minor stylistic comment, but I would remove the italics for above and below ground throughout.

We have removed italics from above and below ground and agree that it improves the manuscript.

Section 2.1: Could you add a photo of one of your DTPs to Fig S1? Please also give an indication of how deep into the soil these profilers go, and when they were deployed relative to the start of the snow season.

Thank you for your suggestion. We have added to the Supplemental section Figure A2 showing the DTP and iButton setup, and we provide pictures of both instruments in this figure (see below). The DTP sensors measure temperature down to 100 cm to 160 cm depending on the DTP design. We have incorporated this information into the figure caption below. DTPs were deployed late September 2021 and snowfall started on October 20, 2021, information which we have also incorporated into the Methods section of the paper.

[Figure]

*Figure A2. Set up of DTP and iButton sensors. Only shallow soil temperature data was used in this study, but the soil DTPs can measure temperature down to 100 to 160 cm of depth.*

Line 46: Please give the precision of the snow depth estimates.

The estimated snow depth has an uncertainty of ±2.5 cm or ±5 cm, depending on the sensor spacing (5 and 10 cm, respectively). We have incorporated this information into the Methods section of the paper.

Line 47: Is the value of the closest temperature sensor used as the value for $T_{SG}$, or is $T_{SG}$ estimated from the sensor temperature using another method (such as a linear extrapolation)?

The value of the closest temperature sensor is used. We do not apply linear interpolation because temperature data collected below soil/moss is affected by insulation from that layer, and therefore incorporating those measurements into our $T_{SG}$ estimate could confound snow depth predictions.

To clarify this, we provided more detail in the revised manuscript:

P2-L[29]: *"We estimated $T_{SG}$ from the temperature sensor closest to the snow-ground interface, which ranged from 1 to 5 cm above the ground surface and thus avoided impacts of soil or moss on the $T_{SG}$ estimate."*

Line 76-77: Does "shallow subsurface" refer to the 1 - 5 cm temperature measurements from the previous sentence? Consider rephrasing these two sentences for clarity.

Yes, it does. However, we agree that how we phrased it was unclear. Thank you for catching this. We have rephrased these sentences in the revised manuscript.

P2-L[31]: "***Additionally, we extracted shallow subsurface temperature measurements recorded 1 to 5 cm below the ground surface from soil DTPs deployed into the ground.***"

P3-L[29]: "***We used the same hyperparameters and features as RF-Seward, but calculated features from DTP subsurface temperature measurements recorded 1 to 5 cm below the ground surface.***"

Line 84: Consider adding vegetation type for all sites to table S3 and refer to this after the statement "Vegetation also varied across sites". Vegetation for 2 sites is given in the following text but this info isn't currently in the table, whereas vegetation for other sites is included in the sensor details column.

We have added vegetation type to the table (Table C1 in revision) for all sites where sensors were buried beneath the ground surface. We did this because vegetation type only affects snow ground interface temperature measurements when the sensor is placed beneath that vegetation. We then referred to this table after the line "Vegetation also varied" as you suggested. We also added a description of vegetation at the Teller27 and Kougarok64 sites where the machine learning models were trained:

P2-L[19]: "***Vegetation at Teller27 consisted of mixed sedge-willow-Dryas tundra and mixed shrub-sedge tussock tundra-bog, with some areas of tall willow shrubs (Bennett et al., 2022). Vegetation at Kougarok64 consisted of tussock-lichen tundra, alder savanna, tall willow shrubs in willow-birch tundra, tall alder shrubs in alder shrublands, and rocky areas with birch-ericaceous-lichen and sparse Dryas-lichen dwarf shrub tundra (Bennett et al., 2022; Breen et al., 2020).***"

Lines 92-94: I am confused as to how you trained RF-Deep when you are unable to derive depth estimates for snowpacks deeper than the 1.77m length of the temperature probes. Please clarify what data was used to train the deep model.

We have rewritten our description of RF-Deep. RF-Deep was trained using some of the original DTP training data collected on the Seward Peninsula combined with data available at two sites in Senator Beck Basin, Colorado. This data was not collected using DTPs, but rather collocated snow sonic sensors and temperature sensors at an automated weather station. We only used "some" of the original DTP training data because we wanted to balance the dataset such that deeper snow represented a reasonable proportion of the training data (10 %) to reflect the distribution of snow depths at the sites in Colorado. If we included the entire DTP training dataset, we worried that the model would remain biased low, as any snow depths above 1.77 m would reflect a very small percentage of data points. Our updated description of these methods is given below:

P4-L[13]: "***The training data at our study sites was limited to a maximum of 1.77 m due to the length of DTP probes, and thus RF-Seward and RF-Below cannot predict depths***

*greater than 1.77 m. To test if ML could accurately predict deeper snow depths, we trained a third ML model, which we refer to as "RF-Deep". To train this model, we supplemented our original Seward Peninsula training dataset with additional data from two model evaluation sites in Senator Beck Basin, CO, USA with deeper snowpacks (Table C1). The model was applied to one site and trained with data from the other (in addition to the Seward Peninsula DTP data). To mimic the distribution of snow depths at these sites, we ensured that 10 % of the training data consisted of snow depths above 2 m. This reduced the training dataset size compared to other models (Table B2)."*

Line 95: I would role this section into the previous one.

Thank you for this suggestion. We have made this change in the revised manuscript.

Line 115/Figure 1: My initial thought was that "temperature range" referred to the range of temperature measured along the whole depth of the DTP, not just as the snow:ground interface. Consider changing this to "daily $T_{SG}$ range" in both line 115 and the red y-axis for plots g - i. Additionally, the use of blue and green to distinguish between the two different models is not accessible to those with colour vision deficiencies. Please change one of these colours – something like blue and orange or green and purple would work.

Thank you for these edits for Figure 1. We have made the suggested changes and updated the color scheme. See below:

[Figure]

*"Figure 1. Performance of RF-Seward a) evaluated using test data, b) when trained at Teller27 and tested at Kougarok64, and c) visa versa. d-f) Same as a-c but for RF-Below. Time series plots of DTP snow depth data vs. ML estimates when g) trained at both sites, h) trained at Teller2, and i) trained at Kougarok64. The dotted red line shows daily $T_{SG}$ range, with narrower temperature ranges occurring under deeper snow cover. "Train N" refers to the number of DTP sensors used to train each model."*

Line 138: Could this poor performance for ephemeral snowpacks be improved by including more ephemeral snowpacks in the training dataset?

This is a good question. Because zero-curtains completely decrease temperature variability, they mask the impact of snow depth on snow-ground interface temperature and remove the predictive value of temperature data during that period. Therefore, we expect that even with a larger/ more representative dataset, the model would perform poorly during these periods. However, incorporating new features related to zero-curtain periods could potentially reduce these errors.

To address this question in the manuscript, we add the following lines:

P6-L[21]: *"ZCPs completely dampen $T_{SG}$ variability and therefore uncouple $T_{SG}$ from snow depth. Even given training data more representative of ZCPs, snow depth estimates may remain unreliable during these periods. Incorporating features into the model which indicate the presence of ZCPs may reduce these errors. Further, deploying iButtons at the snow-ground interface (rather than below ground) decreases the number of ZCPs in the temperature data."*

We also reorganized the paragraph to better highlight this message.

Line 149: The insulative capacity of some snowpacks has been shown to be reached at much shallower depths than 1 m (e.g., Slater et al, 2017), particularly in Arctic environments like where the original model was trained. Potentially reconsider the use of this 1m value.

Thank you for sharing this reference with us, after reviewing it, we realized that the 1 m value was too high, as you suggest. Therefore, we have changed this value to 50 cm in the text.

Line 156/Figure 2: The figure caption refers to a colour bar for subplot f), when I think you mean the y-axis for subplot g). Please double check. Some units on the y axes are also needed. Please also clarify what the black lines refer to – measured snow depth? Also, as for the previous figure, the use of blue and green to distinguish between the two different models is not accessible to those with colour vision deficiencies. Please change one of these colours.

Thank you for these suggestions, we have updated the figure and figure caption. See below:

[Figure]

*"Figure 2. ML performance at a) Bayelva Station, Svalbard, Norway; b) Imnaviat Creek, Alaska, USA; c,d) Council, Alaska, USA; e) Samoylov Island, Siberia, Russia; f) Ivotuk, Alaska, USA; g) Los Alamos, New Mexico, USA; h) Grand Mesa, Colorado, USA; and i,j) Senator Beck Basin, Colorado, USA. Locations are shown on a map, with the yellow star indicating the Seward Peninsula of Alaska, where RF-Seward was trained. Black lines show measured snow depth at each site. Y-axis and RMSE values indicate snow depth in*

*meters. f) Note adjusted y-axis for Los Alamos, New Mexico. For this site, we also show RF-Seward and RF-Below predictions when RF-Below was applied above ground and RF-Seward was applied below ground (dotted lines)."*

Figure S1: Please clarify what is meant by WY2023 and WY2022 in the figure caption. Can you also confirm that snow depth data shown in b) and d) is for a different year to the temperature data on which the snow depth model is based. Also see comments for Section 2.1 above.

By WY2023 we meant the 2023 water year (October 1 2022 - September 30 2023). We have clarified this in the figure caption by saying the "2022 - 2023 snow season" instead of WY2023 and the "2021 - 2022 snow season" instead of WY2022.

The background snow depth imagery shown in b) and d) was collected in the same year (April 2022) as the DTP data on which the model was trained. We hope that by clarifying the date ranges of data collection we have resolved your uncertainty around this. The figure and updated figure caption are shown below for your convenience.

[Figure]

*"Figure A1. Locations of iButton Link Thermochron (DS1921G-F5#) temperature sensors deployed in (green circles) and outside (white circles) of shrubs over the 2022 – 2023 snow season at a) Teller27 and c) Kougarok64. Background imagery from Esri, Garmin,*

*USGS, Maxar, 2024, ArcGIS RGB Basemap. Locations of DTP temperature sensors that recorded both above and below ground temperature (yellow triangles) or only above ground temperature (red circles) over the 2021 – 2022 snow season at b) Teller27 and d) Kougarok64. Blue background imagery shows snow depth in April 2022 estimated using Light Detection and Ranging (LiDAR) data (Singhania et al., 2023b, a)."*

References:

Slater, A.G., Lawrence, D.M. and Koven, C.D. (2017) 'Process-level model evaluation: a snow and heat transfer metric', *The Cryosphere*, 11 (2), 89–996. https://doi.org/10.5194/tc-11-989-2017.

Citation: https://doi.org/10.5194/egusphere-2024-2249-RC1

---

## Author Comment (AC2)

**Reviewer 2**

This brief communication presents an interesting approach to derive snow depth through temperature data recorded with easy-to-deploy sensors. Authors exploit machine learning models (random forest) to predict snow depth from snow-soil interface temperature. While the brief communication reads well and it is suitable to be published in The Cryosphere, some points must be addressed before publication.

Thank you for your review of our manuscript. We appreciate the time and thought that you have put into your comments.

First of all, the approach tested is trained in two sites and then evaluated in these two sites, but also in 10 other sites, what I might highlight in both the abstract and the introduction.

Per your suggestion, we have highlighted this point in the introduction of the revised manuscript:

P2-L[8]: "***The model was trained at two small sites on the Seward Peninsula, Alaska, USA, and evaluated at ten sites distributed across Alaska, Colorado, and New Mexico (USA), Svalbard (Norway), and Siberia (Russia).***"

The abstract is limited to 100 words so we did not have space to highlight this point there.

Through this test on model transferability it is clear that this approach works well in cold and high latitude areas, but in temperate areas where ROS events can occur or temperatures are milder, it fails.  This has to be highlighted in the abstract and the conclusions.

We agree with your suggestion that the shortcomings of this approach should be highlighted earlier on. One thing to note is that some issues related to zero-curtains and warm, ephemeral snowpacks can largely be avoided when using temperature collected at the snow-ground interface. This is because below ground sensors are impacted by the soil freeze-thaw cycle, whereas the above ground sensors are not. In the abstract and introduction, we highlight results from RF-Seward, and therefore do not discuss the additional shortcomings of using RF-Below.

We added text to the abstract to highlight the shortcomings of our model at temperate sites:

P1-L[17] ***"It performed poorly at temperate sites with deeper snowpacks, partially due to training data limitations."***

We were unable to mention the ROS limitation in the abstract because we are limited to 100 words, but that shortcoming is highlighted in the conclusions.  The full description of shortcomings in the conclusion now reads:

P9-L[19]: "***While the model generally performed well, rain-on-snow events and zero-curtain periods cause the model to erroneously predict snow accumulation events. Further, the model failed to replicate deep snow depths (greater than 1.5 m) observed in Colorado, USA. For optimal performance, the model should be applied to temperatures***

*recorded at the snow-ground interface. Predictions made using temperatures recorded below the ground surface were impacted by varying soil types, vegetation properties, and latent heat processes."*

More details about the training study sites (spatial distribution of DTP's within the domain), image of the DTPs, and photograph of them would be desirable. I guess number of figures are limited, but some of these can be included in Figure 1.

A map of DTP locations was provided in the supplemental material and is copied below for your convenience:

[Figure]

*"Figure A1. Locations of iButton Link Thermochron (DS1921G-F5#) temperature sensors deployed in (green circles) and outside (white circles) of shrubs over the 2022 – 2023 snow season at a) Teller27 and c) Kougarok64. Background imagery from Esri, Garmin, USGS, Maxar, 2024, ArcGIS RGB Basemap. Locations of DTP temperature sensors that recorded both above and below ground temperature (yellow triangles) or only above ground temperature (red circles) over the 2021 – 2022 snow season at b) Teller27 and d) Kougarok64. Blue background imagery shows snow depth in April 2022 estimated using Light Detection and Ranging (LiDAR) data (Singhania et al., 2023b, a)."*

We also introduced an additional figure to the supplemental material (Figure A2) to provide more information on the DTP sensors:

[Figure]

*"Figure A2. Set up of DTP and iButton sensors."*

The application of the training, validation and evaluation datasets it is not clear. This point has to be clarified in methods section.

We have added a figure to the supplemental material (Figure B1) to visually show how the training, validation, and test datasets are applied during model development. The validation dataset is used to test how different feature combinations impact model performance. This way, none of the test dataset is used to inform the development of the final model. The new figure is shown below and referred to in the updated methods section.

[Figure]

*"Figure B1. Use of training, validation, and test datasets in model development. We split the training data into groups of DTPs rather than groups of daily data points to maintain the independence of entire snow depth/ temperature time series during model testing. Different combinations of input features were tested using the validation dataset. After the best-performing set of input features was determined, the final model was trained using both the training dataset and validation dataset. The test dataset was excluded completely from the model development process."*

Similarly, it is not clear if, for sites where the models are transferred, these are evaluated with a similar dataset of observation (DTPs spatially distributed) or just data compared with automatic weather station data from a single location.

Model predictions at these sites are compared with snow depth data collected at a single location within 5 m of the temperature data. Snow depth predictions were recorded using sonic sensors, except for at a site in New Mexico where we manually recorded snow depth. To clarify that we are using individual snow depth measurements and not measurements averaged over a network of sensors, we add an additional sentence below:

P4-L[3]: *"Further, we applied RF-Seward and RF-Below to ten evaluation datasets where $T_{SG}$ and snow depth measurements were collocated (within approximately 5 m of each other). Sites were located in the United States (Alaska, Colorado, and New Mexico), Norway (Svalbard) and Russia (Siberia), with temperature sensors placed at the snow-ground interface or within the top 5 cm of soil (see Table C1). Snow depth was also recorded at the sites (e.g., snow sonic sensors at automated weather stations), and was used to evaluate model performance. "*

Minor comments

Line 30: I assume you already know somehow the spatial distribution of the snowpack in the study area (lidar/uav data?) or you are just modeling and testing in the exact location of your DTP sensors? I think it is the second but it is not clear.

The depth and density values presented here were from end-of-winter snow surveys conducted at the study sites. We clarified this in the text:

P2-L[16]: *"According to end-of-winter snow surveys, the average peak snow depth from 2017-2019 at Teller27 was 0.96 m, with an average density of 310 kg/m$^3$ (Bennett et al., 2022). In 2018, snow depth was shallower at Kougarok64 than at Teller27, with an average end-of-winter depth of 0.75 m and density of 290 kg/m$^3$ (Bennett et al., 2022)."*

Line 38 and 39: Please include snow density units in the international system (Kg/m3).

We have corrected this throughout the manuscript and supplemental material.

Line 9. There are some works which have already exploited random forest to analyze, and simulate snow distribution, showing suitable performances. You might cite here: Meloche et al., 2022 (https://doi.org/10.1002/hyp.14546), Revuelto et al., 2020 (https://doi.org/10.1002/hyp.13951) and Hsu et al., 2024 (https://doi.org/10.31223/X57391)

We agree that these studies are relevant to our research. Bennett et al. (2022) developed a random forest machine learning model to predict peak SWE at our study site. Because we are limited on the number of citations we can include, we chose to cite the Bennett et al. (2022) study in our introduction as it is most relevant to our paper:

P2-L[3]: *"Machine learning (ML) models can be used to extrapolate snow survey data, but these estimates still only represent a single point in time (Bennett et al., 2022)."*

Bennett, K. E., Miller, G., Busey, R., Chen, M., Lathrop, E. R., Dann, J. B., Nutt, M., Crumley, R., Dillard, S. L., Dafflon, B., Kumar, J., Bolton, W. R., Wilson, C. J., Iversen, C. M., and Wullschleger, S. D.: Spatial patterns of snow distribution in the sub-Arctic, The Cryosphere, 16, 3269–3293, https://doi.org/10.5194/tc-16-3269-2022, 2022.

Line 69-70: Did you apply an "out of the bag" approach to validate evaluate? I do not understand why you use a 24 DTP validation data and a 31 DTP evaluation dataset, which is the difference here? If not, why don't you use an out of the bag test?

We did not use an "out of the bag" approach because the model is trained using daily data, and we wanted to hold out entire sensors for validation/testing rather than individual daily data points. We suspected that if we held out individual (daily) data points (as done in an "out of the bag" approach), our error estimates would underestimate model error, as the model likely would have seen similar data from neighboring days recorded using the same DTP sensor during model training. By holding out entire sensors, we hoped that our error estimates would be more realistic.

To clarify why we chose to split our training/validation/test datasets into groups of sensors, we add the following sentence to the caption for Figure B1:

*"We split the training data into groups of DTPs rather than groups of daily data points to maintain the independence of entire snow depth/ temperature time series during model testing."*

We hope that our addition of Figure B1 helps clarify how we use the training/validation/test tests. Mainly, we use the validation set to evaluate how different combinations of input features impact model performance.

Lines 72-77: Impact of sensor burial. I would present this section on section 2.2.

Thank you for this suggestion. We have made this change in the updated manuscript.

Line 90: How many sensors are used to train RF-Deep in senator Beck Basin? is this a similar test area (i.e. same number of DTPs or equivalent sensors)?

Far fewer training data points were used to train RF-Deep than the other machine learning models. At Senator Beck Basin, there were two automated weather stations which recorded both snow depth and snow-ground interface temperature. We combined this data with the data collected at the Seward Peninsula. We then balanced the combined Seward Peninsula + Senator Beck training dataset such that deeper snow represented a reasonable proportion of the training data (10 %) to reflect the distribution of snow depths at the sites in Colorado. If we included the entire DTP training dataset, we worried that the model would remain biased low, as any snow depths above 1.77 m would reflect a very small percentage of data points. Our updated description of these methods is given below:

P4-L[13]: *"The training data at our study sites was limited to a maximum of 1.77 m due to the length of DTP probes, and thus RF-Seward and RF-Below cannot predict depths greater than 1.77 m. To test if ML could accurately predict deeper snow depths, we trained a third ML model, which we refer to as "RF-Deep". To train this model, we*

*supplemented our original Seward Peninsula training dataset with additional data from two model evaluation sites in Senator Beck Basin, CO, USA with deeper snowpacks (Table C1). The model was applied to one site and trained with data from the other (in addition to the Seward Peninsula DTP data). To mimic the distribution of snow depths at these sites, we ensured that 10 % of the training data consisted of snow depths above 2 m. This reduced the training dataset size compared to other models (Table B2)."*

We also added Table B2 in the supplemental material to show how many training data points were used to train each machine learning model:

| Model | Number of training data points | Related figure |
|---|---|---|
| RF-Seward (applied to the test dataset) | 20,963 | 1a |
| RF-Seward (trained at Teller27 and tested at Kougarok64) | 17,171 | 1b |
| RF-Seward (trained at Kougarok64 and tested at Teller27) | 9,272 | 1c |
| RF-Seward (retrained on all DTP data; applied to evaluation sites) | 25,418 | 2a-g, h |
| RF-Below (applied to the test dataset) | 15,197 | 1d |
| RF-Below (trained at Teller27 and tested at Kougarok64) | 11,396 | 1e |
| RF-Below (trained at Kougarok64 and tested at Teller27) | 7,980 | 1f |
| RF-Below (retrained on all DTP data; applied to evaluation sites) | 18,968 | 2c-h |
| RF-Deep (applied to first Senator Beck Basin Site) | 1,305 | 2i |
| RF-Deep (applied to second Senator Beck Basin Site) | 3,294 | 2j |

**Table B2. Number of training data points (days) used to train the random forest models.**

Line 114. I would briefly state here how do you test these models. You are directly comparing the observed snow depth at the sensor location in different stations with that modeled, right?

Thank you for this suggestion. We are testing these models by comparing them to snow depth measured at the site. We have clarified this in the methods:

P4-L[6]: "*Snow depth was also recorded at the sites (e.g., snow sonic sensors at automated weather stations), and was used to evaluate model performance.*"

Figure 2. Some symbols of the study area are quite difficult to identify (eg. Bayleva station or Siberian), please increase their size. Also captions and graphs sizes are too small, can this figure be extended and increase captions size. For instance, you can remove the names above the graphs and just include the letter inside each one (a), b), c),…).

We have made the caption and graph sizes larger as you suggested. We also added letters (a,b,c, etc.) to the site map to make the symbols easier to identify and pair with their corresponding time series plot. The updated figure is shown below:

[Figure]

*"Figure 2. ML performance at a) Bayelva Station, Svalbard, Norway; b) Imnaviat Creek, Alaska, USA; c,d) Council, Alaska, USA; e) Samoylov Island, Siberia, Russia; f) Ivotuk, Alaska, USA; g) Los Alamos, New Mexico, USA; h) Grand Mesa, Colorado, USA; and i,j) Senator Beck Basin, Colorado, USA. Locations are shown on a map, with the yellow star indicating the Seward Peninsula of Alaska, where RF-Seward was trained. Black lines show measured snow depth at each site. Y-axis and RMSE values indicate snow depth in*

*meters. f) Note adjusted y-axis for Los Alamos, New Mexico. For this site, we also show RF-Seward and RF-Below predictions when RF-Below was applied above ground and RF-Seward was applied belowground (dotted lines)."*

Conclusions: It must be highlighted that this method is suitable to predict snow depth in cold regions and that its applicability in temperate areas must be further investigated.

Thank you for your review of our manuscript. We hope that our additions to the abstract, introduction, and conclusions help highlight this point.

Citation: https://doi.org/10.5194/egusphere-2024-2249-RC2

---

## Author Comment (AC3)

**Reviewer 3**

Summary

This brief communication describes a new method to use inexpensive temperature sensors and machine learning to estimate snow depth in the Arctic, with cross-validation in temperate regions. The manuscript presents the results clearly and succinctly, and my only major comments relate to the presentation of information, rather than the analyses conducted. I recommend that this manuscript be published following minor revisions.

Thank you for your review of our manuscript. We appreciate the time and thought that you have put into your comments.

Major comments

The non-Arctic sites should be introduced somewhere in the methods – as it is, they come as a bit of a surprise in the results, making it difficult to track what data are used and how.

Thank you for your suggestion. We have introduced these sites earlier in the manuscript now.

Our abstract is very limited on space (100 words maximum), but we have added the following sentence:

P1-L[17]: "*It performs poorly at temperate sites with deeper snowpacks, partially due to training data limitations.*"

 We have also added text to the introduction:

P2-L[8]:  "*The model was trained at two small sites on the Seward Peninsula, Alaska, USA, and evaluated at ten sites distributed across Alaska, Colorado, and New Mexico (USA), Svalbard (Norway), and Siberia (Russia).*"

These sites are also mentioned in the methods section:

P4-L[3]: "*Further, we applied RF-Seward and RF-Below to ten evaluation datasets where $T_{SG}$ and snow depth measurements were collocated (within approximately 5 m of each other). Sites were located in the United States (Alaska, Colorado, and New Mexico), Norway (Svalbard) and Russia (Siberia), with temperature sensors placed at the snow-ground interface or within the top 5 cm of soil (see Table C1).*"

We hope that these sites come as less of a surprise now that they are mentioned in the abstract and introduction.

I'm sure space is short, but I worry that the description in the abstract noting that the model performed "well" is a little bit misleading, as the RMSE = 0.15 m is among the lowest you report, and whether or not that should be considered good performance is a matter of judgement. I'd

like to see a little more nuance in the abstract – maybe a brief description of the conditions under which the model performs best and worst, with the relevant RMSE values provided.

Our abstract is limited to 100 words, but we have added that the model performs poorly at temperate sites (e.g., Colorado). We hope that our changes to the abstract clarify that the statistic of RMSE = 0.15 m only holds true for Arctic sites:

P1-L[16]: *"The model performed well on Alaska's Seward Peninsula where it was trained, and at Arctic evaluation sites (RMSE  0.15 m). It performed poorly at temperate sites with deeper snowpacks, partially due to training data limitations."*

A full description of model limitations is provided in the conclusions.

Percent bias could also be helpful here, given that snow depth is so important to model performance.

We chose to present an RMSE value rather than a percent bias value in the abstract because we compare model performance across sites, and RMSE is directly comparable between sites whereas percent bias is not. For example, small errors at a site with low snow depths would likely result in high percent errors, even though the magnitude of errors is small.

However, we have added mean bias values to our results section (see P4-L[27-29], for example).

Minor comments

Line 19 – citation needed here, as this probably refers mainly to potential for increasing snow depth?

Thank you for noting this. We agree, and we have added the following citations:

Bigalke, S. and Walsh, J. E.: Future Changes of Snow in Alaska and the Arctic under Stabilized Global Warming Scenarios, Atmosphere, 13, 541, https://doi.org/10.3390/atmos13040541, 2022.

Pedron, S. A., Jespersen, R. G., Xu, X., Khazindar, Y., Welker, J. M., and Czimczik, C. I.: More Snow Accelerates Legacy Carbon Emissions From Arctic Permafrost, AGU Advances, 4, e2023AV000942, https://doi.org/10.1029/2023AV000942, 2023.

Line 25-26 – Sonic sensors are deployed at SNOTEL stations, along with snow pillows, but this currently reads as though sonic sensors and SNOTEL stations are two distinct types of monitoring equipment. Suggest rewording.

You are right that the wording was misleading. We have reworded this sentence:

**P2-L[5]:** *"The temporal evolution of snow can be monitored using automated instruments (e.g., snow sonic sensors deployed at Snow Telemetry (SNOTEL) stations; Fleming et al., 2023), but spatially distributed deployment is time consuming and expensive. "*

Line 28 – Can you say why these remain a challenge in Arctic regions? In fact, I would expect IceSat-2 to provide better observations in polar than temperate regions, due to the higher sampling density.

Arctic snow depths vary across very fine spatial and temporal scales. This is partially because winds in many Arctic regions are very high, so snow blows across the landscape and redistributes quickly, which creates a patterned landscape of drifted and scoured areas. Satellites cannot capture those fine scale patterns because they operate at relatively coarse temporal and spatial scales. It is important to actually measure the fine-scale variations in Arctic snowpack because drifts may impact (warm) permafrost. For example, as shrubs expand, it is likely that where drifts form on the landscape will change.

We tied this reasoning into our introduction by adding a few sentences:

**P1-L[26]:** *"As shrubs expand in the Arctic ( Mekonnen et al., 2021), the spatial distribution of snow drifts and subsequent impacts on permafrost may change (Lathrop et al., 2024). Thus, monitoring and modeling fine-scale drifting processes are crucial to understanding permafrost evolution"*

**P2-L[1]:** "*Satellite data can be used to estimate snow depth (Besso et al., 2024), but spatial and temporal resolutions are too coarse to capture the complexity of Arctic snowpacks."*

Line 59 – I don't think this permutation importance is unique to RF; should remove as a reason for selecting RF. Your other reasons for selecting RF are perfectly good, though.

Thank you for catching this, we have rephrased those sentences. We still say that random forests are easier to interpret than other models because more feature importance metrics exist for random forests (e.g. gini importance) and individual trees can be examined to understand how the model is making its decisions.

The updated text is shown below:

**P3-L[12]:** *"We chose a random forest as it outperformed or performed similarly to other models. A random forest is simple to design, computationally inexpensive, and easy to interpret. We identified key model features using permutation importance, which reflects how model performance changes when an input feature is randomly shuffled (Breiman, 2001). Larger decreases in performances indicate greater feature importance."*

Line 90 – I think this is the first time the other training sites are being introduced. They should be briefly described somewhere.

We have now introduced these sites in the abstract and introduction (see previous response).

Line 134 – I think you should define the zero-curtain period the first time you use the term.

Thank you for this suggestion. We first use the term "zero-curtain" when discussing Figure 1. We have updated the manuscript text to provide a more detailed description of a zero-curtain period:

P5-L[4]: *"Further, warmer and/or wetter sites (e.g., Teller27) undergo more freezing and thawing than colder and/or dryer sites (e.g., Kougarok64), producing zero-curtain periods where the key snow depth predictor (temperature variability) flattens at 0°C as water changes phase (Staub and Delaloye, 2017)."*

Line 180-181 – I question whether future work should try to improve the technique for deeper snow – it seems that for physical reasons, this may be unlikely. Perhaps it would be more productive to discuss how the technique could be combined with other types of observations.

We agree that it is possible that this technique will never work for deep snow. However, the dataset used in this study had no deep snow estimates in it at all, so the model could not possibly predict deep snow even if some relationship with depth and temperature still existed. Because of this, we think it is worth exploring whether this technique works given a more representative training dataset. We did try to test this using "RF-Deep", but the data used to train that dataset was not as high quality as what we used to train RF-Seward and RF-Below, and we used far fewer training data points. We have provided a more nuanced discussion of this in the conclusions.

We also agree that this technique could be combined with observations/models to improve estimates even in regions where deep snow limits model performance. We have added some brief discussion around this in the conclusions as well. See below:

P9-L[26]: "*Future research should focus on developing this technique for locations where peak snow depths exceed 1.5 m (e.g., Colorado, USA), as these regions are crucial for water security across the world. While deep snow may completely dampen $T_{SG}$, it is possible that the ML model will perform better given a larger and more representative training dataset and/or additional input features. Alternatively, this technique could be combined with other monitoring and/or modeling efforts. For example, snow depth estimates made early in the snow season (e.g. when snow is shallow) could be used to estimate snow variability across the landscape and to downscale coarse model or remote sensing snow depth estimates.*"

I also wonder about discussing a more thorough investigation of the relative merits of different ML models; an LSTM would make more sense conceptually but is probably harder to implement, and we're not given much information about the implementation you tried that didn't outperform the RF.

This is a great suggestion. LSTMs have the potential for modeling snow dynamics given their ability to capture temporal dependencies. However, they require sufficient data to learn these relationships. The lack of a complete snow cycle likely hindered the LSTM's ability to effectively learn the seasonal patterns. Additionally, there is a trade off between having more training samples with a shorter look-back window and having less samples with a longer look-back window. With a longer dataset encompassing multiple years, we anticipate that an LSTM could potentially improve performance. We summarized this briefly in the conclusions:

P9-L[31]: *"Further, the application of a ML model tailored towards time series estimates (e.g., a Long Short Term Memory Model; LSTM) could improve predictions. In this study, we only had one year of data, which likely limited the LSTM's performance. With a longer-term dataset, we could provide the LSTM with more training points and a longer look-back window (e.g., an entire snow season), which would likely enhance its performance."*

Citation: https://doi.org/10.5194/egusphere-2024-2249-RC3